# Anti-racist and anti-colonial content within US global health curricula

**Sanemba Aya Fanny**[1,2]*, **Amy Rule**[3], **Heather L. Crouse**[2], **James C. Hudspeth**[4], **Bethany Hodge**[5], **Marideth Rus**[2], **Heather Haq**[6,7]

1 Department of Pediatrics, Division of Pediatric Emergency Medicine, Emory University School of Medicine, Atlanta, Georgia, United States of America, 2 Department of Pediatrics, Division of Pediatric Emergency Medicine, Baylor College of Medicine, Houston, Texas, United States of America, 3 Department of Pediatrics, Divisions of Neonatology and Hospital Medicine, Emory University School of Medicine, Atlanta, Georgia, United States of America, 4 Department of Medicine, Section of General Internal Medicine, Boston University, Boston, Massachusetts, United States of America, 5 Department of Pediatrics, Division of Pediatric Hospital Medicine, University of Louisville School of Medicine, Louisville, Kentucky, United States of America, 6 Baylor College of Medicine International Pediatric AIDS Initiative at Texas Children's Hospital, Houston, Texas, United States of America, 7 Department of Pediatrics, Division of Pediatric Hospital Medicine, Baylor College of Medicine, Houston, Texas, United States of America

* sfanny@emory.edu

## Abstract

There is a growing interest to address pervasive racist and colonialist practices in global health (GH). However, there is a paucity of information on anti-racist and anti-colonial (ARAC) education for GH trainees. This study aimed to identify curricular strengths and gaps in ARAC content for pediatric, family medicine and emergency medicine trainees participating in GH. We conducted a cross-sectional survey of GH programs' ARAC curricular content from May 2021 to January 2022. The survey was distributed to 148 GH program educational leaders via email. Descriptive statistics were used to describe quantitative data and comments were reviewed for common themes. The survey response rate was 44% (n = 65). The most represented programs were pediatric residency GH tracks (n = 24, 37%) and emergency medicine (n = 13, 20%), family medicine (n = 4, 6%) and pediatric emergency medicine (n = 6, 9%) GH fellowships. 28% of programs (n = 18) did not have faculty who identify as underrepresented minorities or international medical graduates. 56 programs (86%) had a formal GH pre-departure curriculum. The following areas were the least covered in respondents' curricula: anti-racism (n = 34, 53%), white saviorism (n = 34, 53%), history of GH (n = 24, 37.5%). 63% (n = 40) had bidirectional exchanges of trainees or faculty, but often with significant limitations. While most GH programs recognized the need for formal pre-departure training prior to international experiences, we identified a lack of diversity among GH faculty, significant areas for improvement in curricular content, and a need for more robust bi-directional partnerships. A more equitable future in GH hinges on addressing these educational gaps.

## Introduction

Over the past three decades, global health (GH) has gained in popularity among medical trainees and professionals across all specialties in the United States of America (USA) [1,2]. In

**Data availability statement:** Our data has been published on the data repository dataverse ans is available here: https://doi.org/10.15139/S3/X3E8A2

**Funding:** The authors received no specific funding for this work.

**Competing interests:** The authors have declared that no competing interests exist.

order to meet this demand, residencies and fellowships across specialties have created a range of GH learning opportunities [1,3–7].

The modern field of GH evolved from colonial, missionary and tropical medicine, which existed primarily to support the colonialist agendas of Western powers [8–11]. After World War II and the rise of investment in public health education, paradigms shifted. In 2008, Koplan and colleagues defined GH as "an area for study, research, and practice that places a priority on improving health and achieving equity in health for all people worldwide" [8]. GH was supposed to usher in a "shift in philosophy and attitude that emphasizes the mutuality of real partnership, a pooling of experience and knowledge, and a two-way flow between developed and developing nations" [8].

Now, more than a decade later, a growing number of stakeholders in the field condemn persistent racist and colonialist practices perpetuating inequities in GH and call for the decolonization of GH. Abimbola and Pai define the decolonization of GH as "the removal of all forms of supremacy - not just white supremacy or male domination but also racism,within all spaces of GH practice, including within countries, between countries, and at the global level" [10–16]. Previous studies have described improvements in GH education in the USA including pre-departure education, clinical ethics and cultural humility [2,4,17]. However, there is a paucity of information on the extent and nature of anti-racist and anti-colonial (ARAC) education for medical trainees who participate in GH. This study aims to identify strengths and gaps in the anti-racist and anti-colonial content from the curriculum of U.S.- based pediatric, family medicine (FM) and emergency medicine (EM) GH training programs.

## Methods

### Survey description

This was a cross-sectional study. The author group of GH educators developed a survey using an iterative process based on literature review [15,17–19] and subject matter expertise including consulting with a large multi-national global health educator group [20,21]. We included questions on program characteristics, GH pre-departure training to enhance knowledge of local context (culture, history, language), longitudinal curriculum topics and teaching modalities, research activities and oversight, trainee evaluation mechanisms, bi-directional partnerships and barriers to implementing ARAC measures at the program level (see S1 questionnaire).

Longitudinal curriculum topics included the following sections, based on prior review of the literature and needs assessment of pediatric global health educators: history of global health (history of colonialism, history of global health), US health disparities: past and present (indigenous health, history of US racial disparities, current US racial health disparities, immigrant and refugee health, health inequity), systemic bias and intersectionality (anti-racism, white saviorism, critical consciousness, cultural humility, examination of motivation for GH engagement, social media, photography ethics), power dynamics in GH electives and partnerships [22].

When inquiring about curricular topics covered, we used the following phrases to describe the emphasis given to each topic: not covered; minimal exposure, mentioned in at least one lecture or activity, not a learning objective; moderately covered, specific learning objective in at least one lecture or activity; and strongly emphasized, both a specific learning objective in two different lectures or activities, and a research activity or written assignment topic.

The survey was administered via Research Electronic Data Capture (REDCap) version 6.10.11 (Vanderbilt University, Nashville, TN, https://www.project-redcap.org/), encrypted, and hosted on a secure server.

### Survey participants

Survey participants were identified through distribution lists from previous surveys of GH pediatric residencies [4] and fellowships [1] based in the United States of America, and publicly available listings of FM and EM GH tracks and fellowships [23,24].

The survey invitation was distributed by electronic mail to 148 pediatric, FM, and EM GH educators (i.e., program directors, assistant program directors or other educational leadership role). The invitation specified that only one person per program should complete the survey. The survey was open between 17/05/2021 and 02/01/2022. Invitations were sent bi-weekly for the first month, and thereafter our author group individually contacted nonresponding potential participants to encourage participation.

### Researcher's reflexivity

Our study aims to identify strengths and gaps in the anti-racist and anti-colonial content from the curriculum of U.S.- based pediatric, family medicine and emergency medicine GH training programs. As such, our author group is based in the United States and conceptualized this study to address U.S.-based programs' research and policy priorities. We acknowledge that our varied life experiences influence our study design and data interpretation in this study. We further acknowledge that although this was a U.S.-based study, the addition of low- and middle-income country-based authors would have enriched our authorship team and the conceptualization, design, and interpretation of the study.

### Ethics

Ethics approval was obtained from the Institutional Review Board at Baylor College of Medicine (H-49440). We received formal written informed consent from all participants.

### Data analysis

Quantitative data were abstracted from REDCap and descriptive statistics were performed using Microsoft Excel (Microsoft Corporation, Washington, USA). Maps were generated using RStudio version 2023.06.2 + 561 (RStudio, PBC, Boston, MA). Median was generally used over mean due to a small number of very large programs. Open text responses were reviewed for common themes using a content analysis approach [25].

## Results

65 out of 148 (44%) targeted GH programs representing at least 40 different institutions responded to the survey. Since this was an anonymous survey, the participants' institution could only be identified through voluntarily provided information linking them to their institution. Thirty were residency programs, 27 were fellowship programs and 8 were either institution-wide programs or programs that included medical students.

### Program characteristics

All Association of Pediatric Program Directors (APPD) regions were represented (Table 1). Residency programs with GH tracks had a median of 5 residents per year versus a median of 1 fellow per year.

All residency programs offered short-term (fewer than 3 months) GH international rotations; 17% (n = 5) and 41% (n = 12) also offered short-term domestic rotations in border health and the Indian Health Services (IHS), respectively. Only one residency program (3%) offered an international rotation for greater than 3 months.

**Table 1. Geographical distribution of programs represented by survey respondents (n = 65).**

| APPD Region* | Number of programs (%) |
|---|---|
| Mid-America Region | 10 (15.4%) |
| Mid-Atlantic Region | 6 (9.2%) |
| Midwest Region | 9 (13.8%) |
| New York Region | 4 (6.2%) |
| Northeast Region | 10 (15.4%) |
| Southeast Region | 9 (13.8%) |
| Southwest Region | 5 (7.7%) |
| Western Region | 12 (18.5%) |

*APPD regions are regions used by the American Association of Pediatric Program Directors to group pediatric training programs into zones. The list of states included in each region is available in S1 questionnaire.

In contrast, most fellowships (n = 25, 93%) offered greater than 3 months of training abroad. Seven fellowships (26%) offered short-term rotations abroad. One fellowship (4%) had both long-term (over 3 months) and short-term IHS training opportunities.

Partnerships were reported in 67 countries, mostly located in South America, Eastern and Southern Africa and Asia (Fig 1). In the USA, programs reported offering "local-global" opportunities in their local communities in Los Angeles, Chicago, Detroit, Wisconsin, Alaska, the Twin Cities, Washington, D.C., the Texas/Mexico border, IHS facilities and American Samoa.

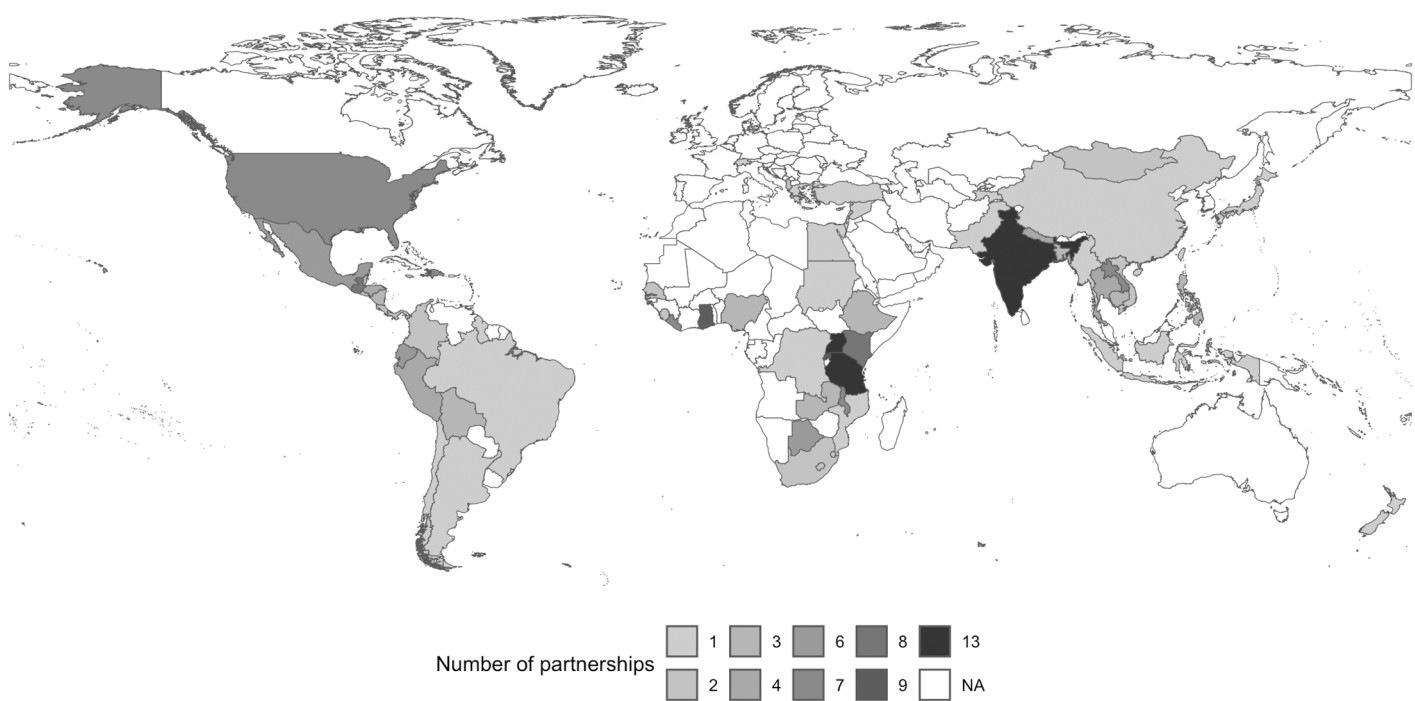

**Fig 1. Geographical distribution of global health partnerships held by programs represented by survey respondents.** Base layer and terms of use of shapefile: https://datacatalog.worldbank.org/search/dataset/0038272/World-Bank-Official-Boundaries. This map is licensed under Creative Commons Attribution 4.0.

Nearly one-third (n = 18, 28%) of GH programs surveyed did not have faculty members who identify as either underrepresented minorities in medicine (URiM) or international medical graduates (IMG). We did not ask the total number of faculty per program.

## Curricular content

Most programs had a formal pre-departure curriculum for learners (n = 56, 86%). Pre-departure requirements varied greatly among programs. While the majority required trainees to learn about their host country's culture (n = 54, 83%) and history and geopolitics (n = 47, 72%), only 54% (n = 35) required language instruction, and 15% (n = 10) did not require any of these topics to be learned prior to departure (Fig 2). Open text responses revealed some respondents felt that those topics were inevitably learned during the GH experience abroad and others found it logistically difficult to develop a pre-departure curriculum.

Key topics covered in longitudinal GH curricula varied among programs (Fig 3). Most put a strong to moderate emphasis on health inequity (n = 58, 91%), examination of one's motivation for GH engagement (n = 56, 88%), cultural humility [26] (n = 53, 84%) and immigrant and refugee health (n = 53, 83%). Less than half of programs put a strong to moderate emphasis on indigenous health (n = 29, 45%), anti-racism (n = 29, 45%), white saviorism [27] (n = 28, 44%) and the history of colonialism (n = 23, 36%).

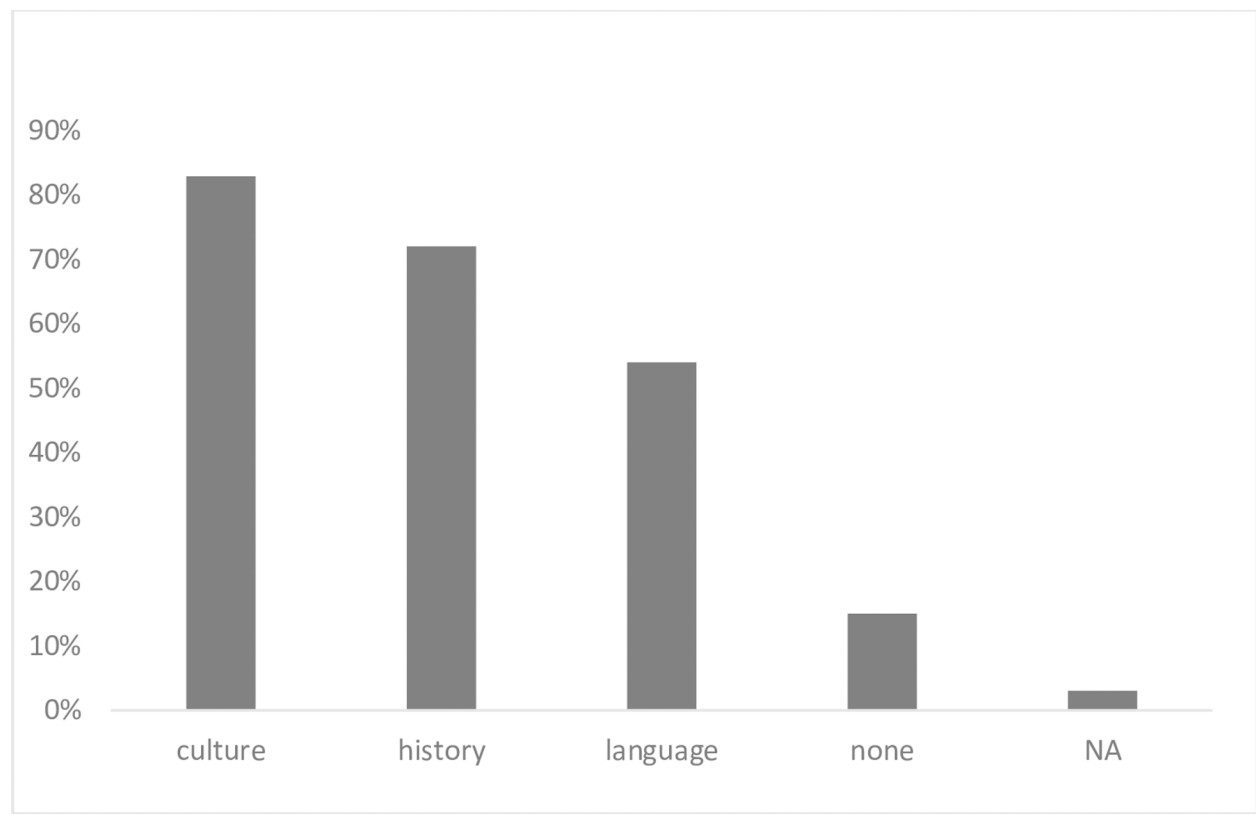

**Fig 2. Pre-departure requirements at US pediatric, emergency medicine and family medicine global health programs (n = 65).**

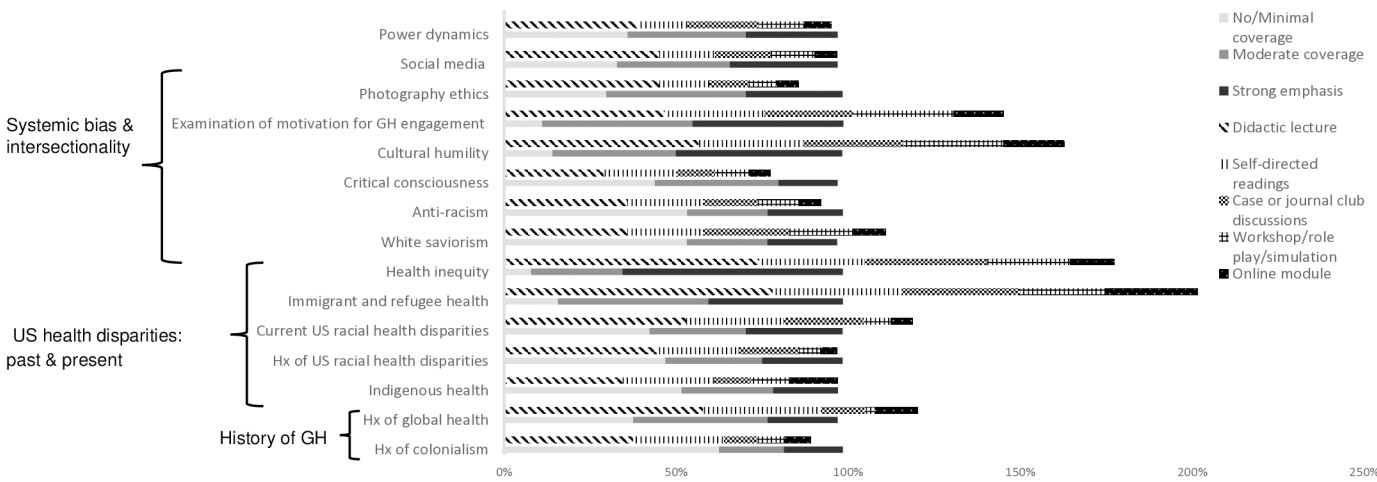

**Fig 3. Curricular contents relating to anti-racism and anti-colonialism covered in global health curricula at programs surveyed and teaching modalities.** Hx = history; US = United States; GH = global health. Please note that teaching modalities can add up to > 100% as it is possible to teach one topic in a variety of ways.

## Procedures and policies

Nearly half of the programs surveyed had written policies on social media practices (n = 28, 44%) and photography ethics (n = 31, 48%). Half had policies addressing practice within one's scope of training while on GH rotations (n = 35, 55%).

## Research

Research was elective for learners at 63% (n = 40) of programs surveyed, while 28% (n = 18) required their learners to participate in research during their GH experience. Most programs that had trainees participate in research required them to engage host site collaborators as co-authors (n = 35, 60%), while 38% (n = 22) encouraged but did not require their learners to do so. Most programs (n = 49, 85%) jointly set the research agenda with their low- and middle-income countries (LMIC) partners. Only 57% (n = 33) always paired their trainees with a local mentor for research. Barriers for LMIC partners participation in setting research objectives and mentoring visiting U.S. learners included: understaffed teams, lack of infrastructure, schedule conflicts/time difference, and language barriers.

## Bidirectional learner exchanges

Nearly two-thirds of programs (n = 40, 63%) hosted trainees from LMICs prior to the COVID-19 pandemic. Yearly, they sent a median of 4 U.S. learners to their LMIC partners for every 3 LMIC visitors they hosted in the U.S.A. Learners from LMICs visiting U.S. institutions participated in clinical observation and a variety of educational activities. Certain U.S. GH programs were only able to host visiting faculty from LMICs. Some LMIC visitors were responsible for covering all the costs associated with their trip to the U.S. Most often, programs obtained a mix of institutional and private funds to sponsor visitors from LMICs. One program planned to re-allocate funds historically dedicated to U.S. trainees to fund more LMIC visitors. Challenges to hosting LMIC visitors at U.S. institutions most often included institutional will and policies, legal and licensing barriers, and funding.

### U.S. learner evaluations

Over half of programs required U.S. trainee evaluations by LMIC faculty (n = 39, 63%), 26% (n = 16) did not, and 11% (n = 7) of respondents did not know whether this requirement existed for their program. Thirty-six programs (58%) had a formal mechanism for LMIC collaborators and mentors to report concerns about visiting trainees. Learners involved in clinical activities were more likely to be formally evaluated than those performing research at the host site.

### Tools needed to integrate anti-racism and anti-colonialism in GH education

When asked what tools would facilitate integration of ARAC content in GH curricula, most respondents requested a formal curriculum. They suggested materials be varied to make the topics more accessible. It was important to them that those materials be free and available online. Many also requested easily accessible standardized learner assessment tools and written policies for social media and photography practices.

## Discussion

This landscape survey describes the current state of ARAC content in GH training programs across the U.S. It revealed a lack of exposure to key ARAC concepts in current GH curricula, a lack of representation among U.S. based GH educators and a need for more bi-directional and equitable clinical and research partnerships.

### Integrating anti-racism and anti-colonialism in GH curricula

Previous studies of residencies with international electives found that approximately two-thirds offered some formal preparation prior to departure [4,28]. In our study, 86% of programs reported having formal pre-departure curricula, indicating that GH educators across specialties and institutions recognize that GH education does not begin and end with the field experience.

Prior studies revealed that GH pre-departure curricula focused largely on biomedical knowledge and skills, and U.S.-based learners' health, physical and psychological safety [2,17,29]. In this study, while 83% and 72% of programs respectively covered host country culture and history in their pre-departure curricula, less than half focused on anti-racism, white saviorism and the history of colonialism.

Similar to previous studies [17,28], didactic or self-directed learning activities were identified as the predominant mode of instruction. However, many authors have advocated for more comprehensive and interactive GH curricula incorporating critical reflection exercises, simulation and case-based approaches [2,17,29]. While existing published curricula cover important topics such as ethics and partnership building [2], GH curricula that delve deeper into the issues of inequity, racism and colonialism and promotes critical reflection, transformative learning and humility are needed.

### Integrating anti-racism and anti-colonialism in GH education leadership

The majority of programs had faculty members who identify as URiM or IMGs, indicating significant progress towards diversity in the GH education community. However, the remaining 28% of programs that did not have faculty members who identify as either URiM or IMGs mirror the lack of diversity among GH leadership described in the 2020 Global Health 50/50 report as well as a 2021 survey of US-based international development and humanitarian

assistance sector organizations [30,31]. We feel that it is important for not just some but all GH programs to have a diverse group of faculty for multiple reasons. First, this partial lack of diversity reflects and recreates the larger power imbalances in GH. Second, the potential lack of plurality among GH faculty and leadership in some programs limits learners' exposure to diverse voices and perspectives. We acknowledge that URiM and IMGs are not the only groups that represent a diverse group of faculty, and our results are somewhat limited as we did not include other groups. For GH to truly address inequities, the root causes of this representation imbalance in GH education need to be urgently investigated and remedied.

## Integrating anti-racism and anti-colonialism in GH partnerships

While many U.S. programs host LMIC learners, the flow of U.S. learners to LMICs typically remains the institutional priority. Our study's ratio of 4:3 U.S. to LMIC learners is likely well above average, reflecting more advanced, better resourced programs that are likely to have already embraced ARAC principles; the dearth of systematic studies makes this hard to assess. Furthermore, international visitors to U.S. medical institutions are currently limited to short non-clinical experiences, which are less valuable than the international clinical work that U.S. learners experience. We echo Hudspeth et al.'s recommendations for institutions to reevaluate the current GH education model and to provide more equitable educational opportunities for LMIC partners [32].

Enabling partners to formally evaluate and give constructive feedback to learners is an important component of equitable partnership building. In our study, only 58% of programs had a formal reporting mechanism in place, in line with the proportion of LMIC partners who reported having the ability to contact a learner's home institution in Cherniak et. al's study [18]. It is our belief that all GH educational programs, in collaboration with their LMIC partners, should implement formal evaluation and reporting mechanisms as they would for any other educational activity performed in the U.S.

## Integrating anti-racism and anti-colonialism at the institutional level

Only recently, most U.S. institutions have created diversity, equity and inclusion (DEI) committees and task forces. GH programming should not be left out of DEI discussions and decisions. GH training programs and their leaders require institutional support to implement meaningful changes towards equity. Institutional support includes policies and practices that reflect the institution's commitment to equity. Examples of policies and best practices include institutional policies that address GH pre-departure training, GH trainee scope of practice, social media practices and photography ethics. Templates for these can be found in the American Board of Pediatrics implementation guide for GH program directors [33]. For programs that offer short-term GH rotations, following the Brocher declaration's ethical principles for short-term GH engagements is another way to commit to more responsible practices [34]. A centralized GH office to monitor compliance and shape the institution's "GH image" (the words and images used by the institution to promote its GH activities) would be particularly helpful for institutions with GH education programs across multiple departments.

Regulatory bodies such as specialty boards and the accreditation council for graduate medical education (ACGME) also have a potential role to play in the ARAC transformation of GH education. For example, although they currently do not provide any oversight over educational GH activities, they could include more rigorous standards such as ensuring LMIC partner participation in all aspects of decision-making and results sharing for research and trainee evaluation for clinical work conducted in a foreign setting as part of specialty training.

## Limitations

Our 44% response rate is the primary limitation of the study, which is potentially multi-factorial. While anonymous, the topics discussed could have led to participant attrition, as could the time required by the detailed questions. Additionally, the period during which the survey was distributed coincided with the COVID-19 pandemic, when many US based GH programs were halted and priorities may have shifted for many educators. Our survey also relied on self-report, and some survey respondents may not have known or disclosed accurate information about their programs. We developed the list of curricular topics through a prior literature review and needs assessment of educators at a conference, and acknowledge that ideally we would have further refined these categories through a Delphi process or similar approach to garner wider input from across our field. While the categories simplify the survey process, they also invariably shape the perspective of survey respondents.

Ideally, our team would have liked to add a qualitative component to further explore barriers to the integration of anti-racism and anti-colonialism in GH curricula and better describe IMG, URiM and LMIC perspectives on GH education. The pandemic and logistical challenges hindered us but our team sees this as a key next step [19]. Too, we did not have LMIC colleagues as part of our authorship group; in hindsight, having the perspectives of our partners who work with the trainees and graduates of these curricula would have helped to refine our study and likely added new insights around our findings, and we will certainly take this step in our further work. Finally, due to logistical limitations (time constraints, inability to obtain reliable contact information for potential participants in other specialties), the survey focused on 3 specialties, and as such, the results may not generalize across GH training, though it is our experience that programs in other fields have similar curricula and approaches.

## Conclusion

The global inequities we face today are the intricate result of centuries of colonialism and white supremacy. It would be naïve to think that they can be swiftly repaired, yet understanding their root causes is paramount to placing equity and justice at the center of global health practice. Anti-racism and anti-colonialism should be better integrated in US GH curricula and at the institutional level. GH educational leadership needs to be diversified. GH partnerships should be bidirectional and have formal evaluation mechanisms. We believe these changes will allow US GH programs to be active partners and allies in anti-racism and decolonization rather than perpetuating colonial inequities..

## Supporting information

**S1 Questionnaire. Questionnaire on anti-racist and anti-colonial content within us global health curricula.**
(PDF)

**S1 Appendix. Author positionality statements.**
(DOCX)

## Acknowledgments

Thank you to Dr. Natalya Kostandova for her technical assistance with producing figures. Above all, we would like to extend our most sincere gratitude to the GHEARD (Global Health Education for Equity, Anti-Racism and Decolonization) authorship group, composed of over 40 authors from 20 institutions and 10 nations around the world. Their collegiality, guidance and feedback during the conceptualization, implementation and interpretation of this

study was invaluable as the first of many steps on our collective journey in the creation of the GHEARD curriculum.

## Author contributions

**Conceptualization:** Sanemba Aya Fanny, Amy Rule, James C. Hudspeth, Bethany Hodge, Marideth Rus, Heather Haq.

**Data curation:** Sanemba Aya Fanny.

**Formal analysis:** Sanemba Aya Fanny.

**Investigation:** Sanemba Aya Fanny.

**Methodology:** Sanemba Aya Fanny, Amy Rule, Heather L. Crouse, James C. Hudspeth, Bethany Hodge, Marideth Rus, Heather Haq.

**Project administration:** Sanemba Aya Fanny.

**Supervision:** Amy Rule, Heather L. Crouse, James C. Hudspeth, Bethany Hodge, Marideth Rus, Heather Haq.

**Visualization:** Sanemba Aya Fanny.

**Writing – original draft:** Sanemba Aya Fanny.

**Writing – review & editing:** Sanemba Aya Fanny, Amy Rule, Heather L. Crouse, James C. Hudspeth, Bethany Hodge, Marideth Rus, Heather Haq.

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
