## [Decision Letter · Decision Letter 0]

20 Jun 2024

PGPH-D-24-01170

Anti-Racist and Anti-Colonial Content within US Global Health Curricula

Dear Dr. Fanny,

Thank you for submitting your manuscript to PLOS Global Public Health. After careful consideration, we feel that it has merit but does not fully meet PLOS Global Public Health’s publication criteria as it currently stands. Therefore, we invite you to submit a revised version of the manuscript that addresses the points raised during the review process.

Thank you so much for this important academic global health article which deals with a very crucial issue of incorporating antiracist and anti-colonial frameworks in to Global Health education. It is an important paper for global health education practitioners to reflect upon, and I would encourage the authors to address comments swiftly so that this gets out into the field as soon as is possible. All changes requested by expert reviewers are essential for the article to get to publication.

In response to the reviewers, kindly include a *reflexivity* statement and a *positionality* statement (either as an appendix or in the manuscript text).

The manuscript should also briefly address why and how particular characteristics chosen (pre-departure training, bidirectional partnerships, trainee evaluation) and why others were not examined. In the training curricula, what is the place of cultural competence and impact assessment with regards to the LMIC host institution? One reviewer has requested strongly that the authors improve categorization of the characteristics that were surveyed preferably based on anti-colonial and anti-racist theories.

A clearer section on implications/call to action with additional extended policy implications will bring more clarity to the study implications.

Both reviewers raise the challenge of greater reflections on bidirectionality in the manuscript (see the comment on emphasis on going to LMICs to teach versus being educated), so that this text is even more strongly decolonial and antiracist.

Kindly address all issues raised by the expert reviewers to strengthen the work and its implications.

With gratitude.

We look forward to receiving your revised manuscript.

Kind regards,

Barnabas Tobi Alayande

Academic Editor

Journal Requirements:

Additional Editor Comments (if provided):

Reviewers' comments:

Reviewer's Responses to Questions

**Comments to the Author**

1. Does this manuscript meet PLOS Global Public Health’s publication criteria ? Is the manuscript technically sound, and do the data support the conclusions? The manuscript must describe methodologically and ethically rigorous research with conclusions that are appropriately drawn based on the data presented.

Reviewer #1: Yes

Reviewer #2: Yes

2. Has the statistical analysis been performed appropriately and rigorously?

Reviewer #1: N/A

Reviewer #2: Yes

3. Have the authors made all data underlying the findings in their manuscript fully available (please refer to the Data Availability Statement at the start of the manuscript PDF file)?

Reviewer #1: Yes

Reviewer #2: Yes

4. Is the manuscript presented in an intelligible fashion and written in standard English?

Reviewer #1: Yes

Reviewer #2: Yes

5. Review Comments to the Author

Reviewer #1: The authors have presented information on anti-racist and anti-colonial content in US global health curricula.

Abstract: The survey was distributed to 148 GH programs. It is not clear if these were program leads? If there are 148 GH programs or these were 148 persons involved in GH? Please clarify because the survey was shared to persons.

As is in methods, line 125 saying the 148 were educators is clearer.

Introduction; Not sure if the [ ] brackets in line 96 were intended. They don't seem serve any purpose.

From the methods, it is not clear if this represents the whole US or these were respondents from a specific state/ geographical area in the US. Would be good in the methods to expound more on the study population. To whom the survey was shared and how it was shared.

remove the word sample in line 142.

Discussion: Line 265 One would say that at least two thirds may represent diversity. The authors could acknowledge this and to maintain their argument probably justify why 100% would be the most ideal.

Probably examine whether the host faculty would be involved with a pre-departure session virtually for trainees to hear from such faculty or the appointment of such faculty as visiting faculty in the US on the program

Line 276 Could also be that the respondents to the survey were the group that is already converted and enthusiastic to see reforms and therefore would support activities/ surveys whose goal is to reform GH

Line 297/298 could come earlier in the paragraph to show how anti racism and anti colonialism were examined at institutional level so that the section is seen to fit with study results.

Line 299 not sure how appropriate it is to provide a web page since this was not a deliverable from this work. The authors could instead provide it as a reference so readers could check out the reference if they need the tool.

Some reflection

Under bi directionality, the curriculum for the trainees from LMICs to US could be analysed as well and then the 2 compared to understand which group (US trainees vs LMIC trainees) benefits most.

In the discussion and in looking at racism, the authors could make comment on the presence or lack of similar curricular in countries that were colonised. About directionality, the curricula sometimes encourages trainees to go teach at international sites with less emphasis on their need to learn from these communities. However, the partners come over to the US as learners and the idea that these can also teach something in the US is lacking. And whether the LMIC trainees can also conduct a research project in the US yet the US Trainees conduct research in LMICs.

Reviewer #2: The paper is a survey of global health curriculum within the training program of several specialties in the US (including paediatric medicine, family medicine, and emergency medicine). The research is well conducted and the paper is well written. The results are interesting and informative. However, there a several flaws that require major revision.

Major

Introduction

Advise the author to outline their positionality and reflexivity in relation to this research. The author correctly identified a lack of representation and diversity in the Faculty of the GH curricula surveyed. What is the author’s positionality? In the methods section, the authors mentioned their ‘subject matter expertise’. What is their subject matter expertise? How is this expertise defined and assessed? Are LMIC partners involved in the assessment of US GH programs? Advise the authors to include a positionality and reflexivity statement.

Methods

The method by which the survey was developed is not entirely clear. In global health, we too often focus on the research that is conducted but not why it is conducted. Why were these particular characteristics chosen (pre-dpearture training, bidirectional partnerships, trainee evaluation)? How were they chosen? Which characteristics are not examined? All these aspects require more attention. It is good that the authors included pre-departure training of the host country’s culture, history, geopolitics, and language, history of colonialism, and etc. Cultural competence appears to be a glaring component of the training curricula that is missing. Another area that is lacking is impact assessment – assessment of the impact of trainee placements by LMIC institutions in terms of impact on the host facility, trainees, and patient care.

It would be better if characteristics surveyed were divided into different sections, e.g. knowledge of global inequities (hx of colonialism), knowledge of local context (history, culture, language), among others. Currently, the categories appear haphazardly selected and arranged.

The survey design would benefit from better theoretical integration of decolonial and anti-racist theories.

Discussion

The discussion is interesting and well-written. It should be better if the call to action or the implications of the study are more clearly highlighted. What is the change that the authors wish to see? What is the purpose of the study? The authors highlighted implications under each subheading. However, it will be nice if they could be tied together at the end. E.g. ‘Anti-racism and anticolonialism should be better integrated in US GH curricula at the institutional level. GH educational leadership needs to be diversified. GH partnerships should be bidirectional and have formal evaluation mechanisms. This will allow US GH programs to be an active form in anti-racism and decolonisation rather than perpetuating colonial inequities, etc’

Beyond having institutional policies on GH programs and short-term GH engagement. Are there other policy implications from this study? Are there standards that specialty training boards can reinforce for training program accreditation?

Minor

abstract

Line 53. Delete the word ‘within’ in this sentence: ‘this study aimed to identify curricular strengths and gaps in ARAC content within for pediatric, family medicine and emergency medicine trainees participating in GH.’

6. PLOS authors have the option to publish the peer review history of their article (what does this mean? ). If published, this will include your full peer review and any attached files.

**Do you want your identity to be public for this peer review?** For information about this choice, including consent withdrawal, please see our Privacy Policy .

Reviewer #1: No

Reviewer #2: No

---

## [Decision Letter · Decision Letter 1]

1 Oct 2024

PGPH-D-24-01170R1

Anti-Racist and Anti-Colonial Content within US Global Health Curricula

Dear Dr. Fanny,

Thank you for submitting your manuscript to PLOS Global Public Health. After careful consideration, we feel that it has merit but does not fully meet PLOS Global Public Health’s publication criteria as it currently stands. Therefore, we invite you to submit a revised version of the manuscript that addresses the points raised during the review process.

Thank you for sending your updated work in based on the initial reviews, and for the  the clarity of responses. One reviewer has made an accept submission, while the other has made a call for a major revision for specific reasons bordering on methodology and authorship inclusion. Many of these issues can be responded to, and can be noted in discussions and limitations- I do not see the feasibility of adjusting survey questionnaires now, or including an LMIC author in a tokenistic gesture. Please respond clearly to the reviewer's concerns and include these notes in limitations where appropriate. This is essential for acceptance.

In responding to the reviewer's comments:

1. The reviewer points out the lack of an LMIC gaze/pose or both. Even though the focus of the paper is on US curricula, the reviewer laments the disadvantage the authors have of the absence of an external view at these curricula. This can easily be couched as a limitation, even if we can argue that those taking a curriculum are in the best position to critique it.

2. Please move the group positionality statement into the main text of the study under methods- perhaps under a heading like "Researcher's Reflexivity". You can leave the individual statements in the supplements.

3. With regards to the ideal categorization of the longitudinal curriculum, and the curriculum in general, this is a retrospective look at what exists, and these categories have already been used in collecting data. Please discuss within limitations that more concise and clear categories could be used reflecting on the reviewer's suggestions.

4. Also respond to the content query around global history and unpack whether this is contained in "history of global health (history of colonialism, history of global health), history of manifest destiny (indigenous health, history of US racial disparities, current US racial health disparities, immigrant and refugee health, health inequity).

Due to it's sensitivity, I would suggest that you replace the history of manifest destiny (which can be misinterpreted for endorsement), even if that was what was originally used, with more direct and clear terminology that perhaps calls out the challenge with the concept of "manifest destiny".

In addition,

5. Please make it clear in the data availability statement that  Dr. Victor Gonzalez, director of research operations for the division of Pediatric Emergency Medicine at Baylor College of Medicine, at vmgonzal@bcm.edu will not be the primary/sole individual handling or determining the availability of this de-identified data. See Plos guidelines and please provide contact information for a data access committee, ethics committee, or other institutional body to which data requests may be sent. "When possible, we recommend authors deposit restricted data to a repository that allows for controlled data access. *If this is not possible, directing data requests to a non-author institutional point of contact, such as a data access or ethics committee, helps guarantee long term stability and availability of data. Providing interested researchers with a durable point of contact ensures data will be accessible even if an author changes email addresses, institutions, or becomes unavailable to answer requests." Please make the committee these requests are going to clear, and provide an institutional/committee email with "attention" contact to address these concerns.

Thank you for this important work.

With gratitude

We look forward to receiving your revised manuscript.

Kind regards,

Barnabas Tobi Alayande

Academic Editor

Journal Requirements:

Additional Editor Comments (if provided):

Thank you for sending your updated work in based on the initial reviews, and for the the clarity of responses. One reviewer has made an accept submission, while the other has made a call for a major revision for specific reasons bordering on methodology and authorship inclusion. Many of these issues can be responded to, and can be noted in discussions and limitations- I do not see the feasibility of adjusting survey questionnaires now, or including an LMIC author in a tokenistic gesture. Please respond clearly to the reviewer's concerns and include these notes in limitations where appropriate.

In responding

1. The reviewer points out the lack of an LMIC gaze/pose or both. Even though the focus of the paper is on US curricula, the reviewer laments the disadvantage the authors have of the absence of an external view at these curricula. This can easily be couched as a limitation, even if we can argue that those taking a curriculum are in the best position to critique it.

2. Please move the group positionality statement into the main text of the study under methods- perhaps under a heading like "Researcher's Reflexivity". You can leave the individual statements in the supplements.

3. With regards to the ideal categorization of the longitudinal curriculum, and the curriculum in general, this is a retrospective look at what exists, and these categories have already been used in collecting data. Please discuss within limitations that more concise and clear categories could be used reflecting on the reviewer's suggestions.

4. Also respond to the content query around global history and unpack whether this is contained in "history of global health (history of colonialism, history of global health), history of manifest destiny (indigenous health, history of US racial disparities, current US racial health disparities, immigrant and refugee health, health inequity).

Due to it's sensitivity, I would suggest that you replace the history of manifest destiny (which can be misinterpreted for endorsement), even if that was what was originally used, with more direct and clear terminology that perhaps calls out the challenge with the concept of "manifest destiny".

In addition,

5. Please make it clear in the data availability statement that Dr. Victor Gonzalez, director of research operations for the division of Pediatric Emergency Medicine at Baylor College of Medicine, at vmgonzal@bcm.edu will not be the primary/sole individual handling or determining the availability of this de-identified data. See Plos guidelines and please provide contact information for a data access committee, ethics committee, or other institutional body to which data requests may be sent. "When possible, we recommend authors deposit restricted data to a repository that allows for controlled data access. *If this is not possible, directing data requests to a non-author institutional point of contact, such as a data access or ethics committee, helps guarantee long term stability and availability of data. Providing interested researchers with a durable point of contact ensures data will be accessible even if an author changes email addresses, institutions, or becomes unavailable to answer requests." Please make the committee these requests are going to clear, and provide an institutional/committee email with "attention" contact to address these concerns.

With gratitude,

Reviewers' comments:

Reviewer's Responses to Questions

**Comments to the Author**

1. If the authors have adequately addressed your comments raised in a previous round of review and you feel that this manuscript is now acceptable for publication, you may indicate that here to bypass the “Comments to the Author” section, enter your conflict of interest statement in the “Confidential to Editor” section, and submit your "Accept" recommendation.

Reviewer #1: All comments have been addressed

Reviewer #2: (No Response)

2. Does this manuscript meet PLOS Global Public Health’s publication criteria ? Is the manuscript technically sound, and do the data support the conclusions? The manuscript must describe methodologically and ethically rigorous research with conclusions that are appropriately drawn based on the data presented.

Reviewer #1: Yes

Reviewer #2: Yes

3. Has the statistical analysis been performed appropriately and rigorously?

Reviewer #1: Yes

Reviewer #2: Yes

4. Have the authors made all data underlying the findings in their manuscript fully available (please refer to the Data Availability Statement at the start of the manuscript PDF file)?

Reviewer #1: Yes

Reviewer #2: No

5. Is the manuscript presented in an intelligible fashion and written in standard English?

Reviewer #1: Yes

Reviewer #2: Yes

6. Review Comments to the Author

Reviewer #1: (No Response)

Reviewer #2: The authors have made improvements to the manuscript.

Major

The lack of LMIC authors is a major limitation in this study. This should be clearly listed as a major limitation in the discussion section of this study. The implications of this limitation should be thoroughly discussed.

It is good that the authors have uploaded their individual and group positionality statements. However, the group positionality statement needs to be included in the main text of the study under introduction or methods and not merely in the appendix.

Line 118 the term ‘history of manifest destiny’ is not the best term to use. It has colonial overtone.

Would be better if the categories are:

History of global health (history of colonialism, history of global health),

Manifestations of inequities (indigenous health, history of US racial disparity, current US racial health disparities, immigrant and refugee health, health inequities)

Systemic bias (anti-racism, white saviourism, critical consciousness).

Intersectionality does not seem to belong here and isn’t explicitly included in your curriculum.

Global health engagement (cultural humility, examination of motivation for GH engagement, social media, photography ethics)

There should be more curriculum content on global history rather than merely US history. Why was this not included in the survey?

7. PLOS authors have the option to publish the peer review history of their article (what does this mean? ). If published, this will include your full peer review and any attached files.

**Do you want your identity to be public for this peer review?** For information about this choice, including consent withdrawal, please see our Privacy Policy .

Reviewer #1: No

Reviewer #2: No

---

## [Decision Letter · Decision Letter 2]

9 Dec 2024

Anti-Racist and Anti-Colonial Content within US Global Health Curricula

PGPH-D-24-01170R2

Dear Dr Fanny,

We are pleased to inform you that your manuscript 'Anti-Racist and Anti-Colonial Content within US Global Health Curricula' has been provisionally accepted for publication in PLOS Global Public Health. Thank you for clearly and concisely addressing all the reviewer and editor's concerns. This is a very important piece of work that adds to our drive for equity in global health engagements from the perspective of the US.

Best regards,

Barnabas Tobi Alayande

Academic Editor

Reviewer Comments (if any, and for reference):

Reviewer's Responses to Questions

**Comments to the Author**

1. If the authors have adequately addressed your comments raised in a previous round of review and you feel that this manuscript is now acceptable for publication, you may indicate that here to bypass the “Comments to the Author” section, enter your conflict of interest statement in the “Confidential to Editor” section, and submit your "Accept" recommendation.

Reviewer #2: All comments have been addressed

2. Does this manuscript meet PLOS Global Public Health’s publication criteria ? Is the manuscript technically sound, and do the data support the conclusions? The manuscript must describe methodologically and ethically rigorous research with conclusions that are appropriately drawn based on the data presented.

Reviewer #2: Yes

3. Has the statistical analysis been performed appropriately and rigorously?

Reviewer #2: N/A

4. Have the authors made all data underlying the findings in their manuscript fully available (please refer to the Data Availability Statement at the start of the manuscript PDF file)?

Reviewer #2: No

5. Is the manuscript presented in an intelligible fashion and written in standard English?

Reviewer #2: Yes

6. Review Comments to the Author

Reviewer #2: (No Response)

7. PLOS authors have the option to publish the peer review history of their article (what does this mean? ). If published, this will include your full peer review and any attached files.

**Do you want your identity to be public for this peer review?** For information about this choice, including consent withdrawal, please see our Privacy Policy .

Reviewer #2: **Yes: ** Rennie Qin
